# Update on the Diagnosis and Management of Medullary Thyroid Cancer: What Has Changed in Recent Years?

**DOI:** 10.3390/cancers14153643

**Published:** 2022-07-27

**Authors:** Krzysztof Kaliszewski, Maksymilian Ludwig, Bartłomiej Ludwig, Agnieszka Mikuła, Maria Greniuk, Jerzy Rudnicki

**Affiliations:** Department of General, Minimally Invasive and Endocrine Surgery, Wroclaw Medical University, Borowska Street 213, 50-556 Wroclaw, Poland; maksymilian.ludwig@student.umw.edu.pl (M.L.); bartlomiej.ludwig@student.umw.edu.pl (B.L.); agnieszka.mikula@student.umw.edu.pl (A.M.); maria.greniuk@student.umw.edu.pl (M.G.); jerzy.rudnicki@umw.edu.pl (J.R.)

**Keywords:** medullary thyroid cancer, laboratory diagnostic, imaging, nuclear medicine, lateral lymph node dissection, transoral thyroidectomy, parathyroid gland identification, multikinase inhibitors, immunotherapy, systematic treatment

## Abstract

**Simple Summary:**

Medullary thyroid carcinoma (MTC) is a rare neoplasm that is responsible for a fair proportion of thyroid carcinoma related deaths. The current diagnostic and therapeutic standards are not always effective and need to be upgraded. The role of biomarkers and immunohistochemistry in the diagnosis of MTC is highlighted. Opportunities for improved diagnostics have been seen with the development of nuclear medicine. Some studies have highlighted the possibility of reducing the number of complications during surgical treatment, which is the basic therapeutic method in patients with MTC. Current pharmacotherapy is imperfect, but there is ongoing research into the use of new, more selective drugs. The following paper discusses recent advances in the diagnosis and treatment of MTC.

**Abstract:**

Medullary thyroid carcinoma (MTC) is a neoplasm originating from parafollicular C cells. MTC is a rare disease, but its prognosis is less favorable than that of well-differentiated thyroid cancers. To improve the prognosis of patients with MTC, early diagnosis and prompt therapeutic management are crucial. In the following paper, recent advances in laboratory and imaging diagnostics and also pharmacological and surgical therapies of MTC are discussed. Currently, a thriving direction of development for laboratory diagnostics is immunohistochemistry. The primary imaging modality in the diagnosis of MTC is the ultrasound, but opportunities for development are seen primarily in nuclear medicine techniques. Surgical management is the primary method of treating MTCs. There are numerous publications concerning the stratification of particular lymph node compartments for removal. With the introduction of more effective methods of intraoperative parathyroid identification, the complication rate of surgical treatment may be reduced. The currently used pharmacotherapy is characterized by high toxicity. Moreover, the main limitation of current pharmacotherapy is the development of drug resistance. Currently, there is ongoing research on the use of tyrosine kinase inhibitors (TKIs), highly specific RET inhibitors, radiotherapy and immunotherapy. These new therapies may improve the prognosis of patients with MTCs.

## 1. Introduction

Thyroid neoplasms are among the most common endocrine pathologies [1]. There are many types of thyroid tumors with different morphological and histological features and also prognoses [2]. The subject of the following paper is medullary thyroid carcinoma (MTC)—specifically, advances in its diagnosis and treatment.

### 1.1. Definition

MTC is a neoplasm originating from parafollicular C cells, which produce calcitonin (Ctn) [2,3]. Ctn is a hormone, which influences calcium homeostasis by inhibiting osteoclasts. In the clinic, Ctn is important in the context of diagnosis and prognosis assessment for patients with MTC [4,5]. The clinical picture is rather nonspecific. One of the most common symptoms is a palpable neck lump. The nodule may cause complaints, such as dysphagia, pain, dyspnea or speech impediment, or it may be asymptomatic. In addition, diarrhea, flushing and chest pain are sometimes observed. Recurrent respiratory infections may be associated with advanced disease [6,7].

### 1.2. Classification

In 2022, the World Health Organization (WHO) proposed the fifth edition of the classification of thyroid cancers. Eight groups of neoplasms were distinguished. Of these, thyroid C-cell-derived carcinomas are a distinct entity, and MTC is its only subgroup [1].

### 1.3. Staging

Table 1 and Table 2 present the tumor, node and metastasis (TNM) staging system according to the Union for International Cancer Control (UICC) [8].

### 1.4. Epidemiology

The incidence of thyroid cancers has increased dramatically over the past 30 years. In 2020, thyroid cancers accounted for 3% of all cancer cases. The main culprit of the described trend is papillary thyroid cancer. In the case of other types, such a clear increase in incidence was not observed [9]. A similar trend was shown by a retrospective population-based study conducted in the United States. It is questionable whether this trend is due to an actual increase in incidence or an increase in cancer detection associated with the development of diagnostic methods [10]. MTC is a rare cancer. Depending on the source, MTC is reported to account for only 1% to 4% of thyroid cancer cases; however, MTC accounts for up to 15% of related deaths [5,6]. Of all MTCs, approximately 25% are familial, and 75% are sporadic [11].

### 1.5. Genetics

MTC can be familial or sporadic. The sporadic form is most often (43–65%) caused by a somatic mutation in the RET (rearranged during transfection) gene, located on the long arm of chromosome 10 (10q11.21). The RET gene is a protooncogene encoding a protein that is part of the receptor for tyrosine kinase. The RET gene is essential for the regulation of cell differentiation, proliferation and apoptosis [7,11]. Mutations involving codons C630 and A883, as well as H568 and S1024, are mainly reported in sporadic cases. In some papers, they are considered as specific to the sporadic form of MTC [12]. However, described cases of germline mutations in both codons C630 and A883 also exist. Their prevalence is, notwithstanding, significantly lower [13]. The second most frequent mutation in sporadic MTCs is a somatic mutation in the RAS gene (in 20–25% of cases), which does not occur in hereditary cases [11]. Rare genetic alterations, also characteristic only of the sporadic form, include small RET deletions or deletions in combination with insertions (most commonly E632-L633del and D898-E901/E902del), multiple co-occurring RET mutations and mutations in multiple genes [12]. Other cases of MTCs are associated with autosomal-dominant inherited familial multiple endocrine neoplasia (MEN) syndromes 2a and 2b or isolated familial medullary thyroid cancer (FMTC) syndrome [11]. MEN 2a occurs in the vast majority of cases (95%) and is distinguished by the following variants: classic pheochromocytoma and parathyroidism, cutaneous lichen amyloidosis and Hirschprung’s disease. In addition to MTC, the clinical presentation of MEN 2b may also include pheochromocytoma, mucosal neuromas and a characteristic (marfanoid) body build [5,14,15]. These syndromes are also caused by mutations in the RET gene, but they occur in germline cells. Only about 2% of cases of familial MTC are not associated with detectable RET germline mutation [16]. We distinguish between intracellular and extracellular RET mutations. MEN 2a is characterized by extracellular domain changes at codons 609, 618, 620 and 634, whereas the M918T mutation in the intracellular domain of tyrosine kinase is detected in MEN 2b [12]. A clear genotype–phenotype correlation is observed in MTCs. For example, the RET M918T mutation is usually associated with poor prognosis. Patients with codon 634 mutations belong to the high-risk category according to the American Thyroid Association (ATA) and may already develop MTC in the first years of life. In contrast, RAS mutations are found in less aggressive cases with a more favorable prognosis [17].

### 1.6. Prognosis

The onset of the disease in sporadic cases is between the ages of 50 and 60 years. Hereditary MTCs usually present earlier [15]. Numerous factors influence prognosis. In addition to the correlation of prognosis with tumor genotype described above, baseline biomarker levels, tumor extent, the presence of metastases (local and distant), sex and age also influence prognosis.

When it comes to biomarkers, routine measurement of basal serum calcitonin (bCtn) is reported as an important part of the diagnostic evaluation [18]. bCtn and carcinoembryonic antigen (CEA) doubling times are considered good tools for assessing prognosis. Based on them, it is possible to classify the patient into a group with stable or progressive disease [19]. The prognosis of mortality in patients with advanced MTC may be assessed with serum Ca9.9 positivity and doubling time [20]. Hajje et al. [21] report that CEA may also be used as a predictive biomarker. Early change in CEA levels is considered as a surrogate marker for progression-free survival (PFS) (*p* = 0.02) in patients with advanced disease, treated with cytotoxic chemotherapy [21].

Histological findings may also be used to assess prognosis. Frank-Raue et al. [22] revealed that poorly differentiated histology is correlated with rapid tumor progression. Apart from routine hematoxylin and eosin staining, immunohistochemical stains were performed as well. They showed that high expression of Ki-67 comes with less favorable prognosis [22].

Raue et al. [23] reviewed whether long-term disease-specific survival (DSS) and outcomes vary between sporadic and hereditary MTCs. They reported that even though hereditary cases usually appear up to 20 years earlier than sporadic ones, they are both characterized by similar tumor aggressiveness [23]. Taking into consideration only the sporadic cases of MTC, it was demonstrated that younger age is associated with longer survival time but not with cancer-related death events [24].

Lymph node metastases are already present at diagnosis in 30–60% of patients. MTCs are associated with a higher mortality rate than well-differentiated thyroid cancers. The 5-year survival rate is estimated at 80–97%, and the 10-year survival rate is estimated at 75–88%. Recurrences occur in up to 50% of patients [5,7].

### 1.7. Diagnosis

A variety of methods, including biochemical, imaging and genetic methods, are used for diagnostic purposes (Table 3). The current guidelines (ATA 2015, ESMO 2019) specify fine-needle aspiration (FNA) biopsy, ultrasound (US) of the neck, serum Ctn and CEA level analyses and RET gene mutation analysis as the most important for MTC detection and management. Depending on the results of the above analyses, a decision is made about the treatment procedure and the search for metastatic foci using contrast-enhanced computed tomography (CT) and magnetic resonance imaging (MRI). The guidelines are not unanimous on the usefulness of nuclear medicine techniques, especially for the detection of secondary foci of MTC [8,25].

### 1.8. Laboratory Diagnostics

According to ATA guidelines, when suspicious nodules are visualized on thyroid ultrasound, FNA biopsy is indicated. US risk stratification system (RSS) was developed to assess the nodules’ character. However, recent studies revealed that US correctly identifies hardly 50% of cases. The reason for such low sensitivity is that the RSS is mostly based on papillary thyroid carcinoma US presentation. That is why a negative US result should not exclude the patient from performing FNA [26]. The sample is cytologically evaluated as part of the standard diagnostic procedure. The histological features of MTCs are salt-and-pepper chromatin, multinucleation and the presence of solid nests of plasmacytoid or spindled cells in a fibrous stroma [6,15,27]. However, the cytological picture can be highly variable. Microscopic MTC tends to be more cohesive and may not have either plasmacytoid or spindled cells. According to recent publications, the sensitivity of FNA biopsy sample analysis can reach up to 86% [6]. Therefore, relying on cytological examination may result in misdiagnosis. In FNA biopsy sample analysis, MTC should be differentiated from follicular neoplasm, sarcoma and plasmacytoma, among others. Another limitation of cytological evaluation may be the low cellularity of the specimen [6,15].

Because of the written limitations of cytological examination, some guidelines recommend routine bCtn testing. There is a discrepancy between the European Thyroid Association (ETA) and ATA recommendations. ETA recommends bCtn measurement in all patients with suspected thyroid nodules. On the other hand, according to ATA recommendations, bCtn measurement should not be performed routinely, mainly for cost-effectiveness reasons. Usually, preoperative diagnosis is based on cytological examination. However, Jassal et al. [28] showed that preoperative bCtn measurement can improve the evaluation of MTCs with indeterminate cytology and be helpful in planning surgery. It has been shown that, in the diagnosis of MTCs, the measurement of serum Ctn concentration has a higher sensitivity than cytological evaluation of FNA biopsy samples, so it is possible that in the future, serum Ctn concentration measurement will be used in routine diagnosis not only in Europe but also elsewhere [29]. Calcitonin concentration is directly proportional to the tumor mass. At bCtn values of 60–100 pg/mL, there is a strong probability of the presence of a C-cell proliferative process. Concentrations above 500 pg/mL may indicate the presence of distant metastases [15]. The main limitation of bCtn is the risk of false-positive results. For example, false-positive results of bCtn measurement may be caused by ectopic production of Ctn by neuroendocrine tumors. Increased serum calcitonin levels are also observed during the use of proton pump inhibitors and in renal failure, pregnancy and hypothyroidism [15,30,31].

To prevent false-positive bCtn results, calcium-stimulated calcitonin (Ca-sCtn) measurement can be used. However, Niederle et al. [30] showed that compared to bCtn measurement, Ca-sCtn measurement does not improve the quality of diagnosis of MTCs. The combination of bCtn and Ca-sCtn testing is also not justified. With the current highly sensitive and specific tests for measuring bCtn, based on immunochemolumetry, the probability of false positives is low [30]. Currently, there are no well-defined cutoff points for bCtn and Ca-sCtn for the diagnosis of MTC. Fugazzola et al. [32] performed an analysis based on which they distinguished the optimal thresholds for separating sick from healthy patients. For bCtn, the optimal thresholds are >30 pg/mL and >34 pg/mL for women and men, respectively, and for Ca-sCtn, they are >79 pg/mL (F) and >466 pg/mL (M). With standardized cutoff points, it is reasonable to consider diagnostic and therapeutic decision making based solely on the concentrations of the described biomarkers [32].

Another marker useful in the laboratory diagnosis of MTCs is CEA, which is produced by C cells. CEA is not specific to medullary thyroid cancer. Increased CEA levels are seen, for example, in smokers and in individuals with inflammatory diseases of the gastrointestinal tract or lung disease. Because of its low specificity, serum CEA level measurement is not suitable as a screening test for the initial evaluation of MTC. However, the amount of secreted CEA increases in proportion to the tumor mass and in the presence of metastases. Therefore, CEA measurement finds application in monitoring the progression of confirmed disease and in staging [6,15,33]. Turkdogan et al. [34] showed that in advanced disease, CEA is an even better prognostic marker than Ctn. Another advantage of CEA measurement is its lower cost than bCtn measurement.

FNA material can also be used for immunohistochemistry. Calcitonin measurement in FNA washout fluids (FNA-Ctn) can be a useful diagnostic technique. Trimboli et al. [35] performed a meta-analysis in which they compared cytology with FNA-Ctn for sensitivity in the diagnosis of MTCs. While the sensitivity of cytology is estimated to be 20–86%, depending on the study, the sensitivity for FNA-Ctn measurement was higher than 95% in most of the publications reviewed. Thus, there is no doubt that FNA-Ctn is a more sensitive diagnostic technique, and it is the method recommended by ATA guidelines. In addition, according to ATA recommendations, when cytology is not conclusive, immunohistochemical staining should also be performed for CEA and chromogranin [25]. There are some limitations of FNA-Ctn. Certain factors may cause a false-positive result. Moreover, they are common to FNA-Ctn and bCtn measurement. For example, C-cell hyperplasia may be such a factor. However, the main limitation of the discussed method is the lack of established cutoff points. Thus, there is a prospect of developing the described technique in laboratory diagnostics of primary MTCs [35]. Marques et al. [36] proposed another application of FNA-Ctn. They evaluated the utility of Ctn measurement in FNA biopsy lymph node material in the diagnosis of metastatic MTCs. The sensitivity and specificity were 81.8% and 97.9% for cytology and 100% and 97.9% for FNA-Ctn, respectively. This shows that FNA-Ctn is a more effective tool for diagnosing the presence of MTC metastasis in a lymph node. The optimal cutoff point was also determined in the described study and was 23.0 pg/mL (*p* < 0.001). This technique has a good chance of development, and it is possible that it will be more widely used in the future [36]. There are numerous less specific biomarkers that show staining in FNA biopsy sample immunohistochemistry with MTC. Thyroid transcription factor-1 (TTF-1) shows diffuse expression not only in tumors originating from the thyroid medulla but also in follicular adenoma, well- and poorly differentiated carcinomas, and C-cell hyperplasia. MTCs are usually characterized by weak and focal expression of paired box gene 8 (PAX8) and thyroid transcription factor-2 (TTF-2). Calcitonin gene-related peptide (α-CGRP), in addition to immunohistochemistry, can also be detected in the serum of patients [37]. Although these markers are not specific to MTC, there is a prospect of the development of immunohistochemistry for these markers in the diagnosis of MTC.

A method that is growing in importance in the laboratory diagnosis of multiple cancers is liquid biopsy. For example, NETest, based on the detection of circulating tumor transcripts, is finding application in the diagnosis of gastrointestinal neuroendocrine tumors (GEP-NENs) [38]. For MTCs, liquid biopsies allow the analysis of circulating microRNAs (miRNAs) and cell-free DNA (cf-DNA) [39].

Galuppini et al. [40] leaned toward the use of circulating miRNAs as biomarkers. They are fragments of 20–22 base pair long, noncoding RNA that play a role in cancer pathogenesis. The function of miRNAs is the post-transcriptional control of gene expression. MiRNAs in blood serum may be potential diagnostic and prognostic biomarkers. Moreover, miRNAs may find application in monitoring the effectiveness of treatment with tyrosine kinase inhibitors (TKIs), which is often not reflected by serum levels of bCtn and CEA. There is some correlation between the expression of specific miRNAs and tumor genotype. The downregulation of miR-224, miR-127 and miR-129-5p and the upregulation of miR-183, miR-153-3p, miR-144 and miR-34a have been demonstrated in RET-positive patients [40]. This knowledge can be helpful in determining disease prognosis.

Ciarletto et al. [41] proposed the use of differential pairwise (diff-pair) analysis of miRNA expression levels for the diagnosis of MTC by FNA biopsy sample analysis. They demonstrated that this test was effective in diagnosing MTCs, even in cases where cytology was not diagnostic. The sensitivity and specificity of diff-pair analysis were 100% and 100%, respectively. Furthermore, there is a chance that based on the expression of specific miRNAs, the prognosis of the disease can be determined. For example, the overexpression of miR-375 and miR-183 correlates with mortality and a tendency toward lymph node metastasis [41]. Further development of miRNA analysis in FNA may significantly improve the diagnosis of MTCs.

Cf-DNA finds application both in the diagnosis of MTCs and in assessing prognosis and response to treatment. In the case of Ctn-negative lesions, cf-DNA works well as a diagnostic biomarker [39]. Cote et al. [42] proved the feasibility of using cf-DNA to detect M918T mutations in MTCs. They showed that a positive test result can provide an alternative to conventional tissue biopsy, but a negative result does not rule out the presence of the mutation. As a positive result for RET M918T cf-DNA is associated with more aggressive disease, opportunities are seen for the use of liquid biopsies in prognosis. The use of cf-DNA as a biomarker of early response to treatment is also suggested [42].

When MTCs are suspected, genetic counseling is indicated. According to ATA and ETA guidelines, tests for RET mutations should be performed in all cases, both sporadic and hereditary, independently of their clinical presentation. When a germline mutation is found, RET genetic screening should be performed in the patient’s relatives. All first-degree family members should be taken into consideration to identify potential subjects with high risk of developing MTC. Their prognosis may be more favorable because of earlier diagnosis and an opportunity to introduce prophylactics treatment [16]. Moreover, RET-positive patients should be monitored for parathyroidism and pheochromocytoma [15,17]. It should be highlighted that when a very rare RET mutation (“variant of unknown significance”, VUS) is found, its role in pathogenesis must be revealed before family testing [16]. In addition to its prognostic value, the detection of RET mutations is important for stratifying patients for treatment with selective TKIs [33].

### 1.9. Morphological Imaging

The first imaging test used in the diagnosis of thyroid tumors is often the ultrasound, as it is a tool that is widely available, inexpensive to use and safe for the patient [8,25]. Moreover, the ATA 2015 guidelines identify the US as the most important preoperative imaging test [25]. MTC on imaging is mostly hypoechoic, has macro- and microcalcifications, is patchy in structure and is vascularized, mainly perinodular. There is rarely a peripheral halo effect [15,43]. It is useful to combine US methods with the measurement of serum Ctn and CEA, which significantly increases the sensitivity of the diagnosis (US vs. US combined with Ctn and CEA—77% vs. 95%) [44].

US can also serve as the first initial tool to evaluate MTC metastasis on regional neck lymph nodes [8,45]. It should be noted that a negative result does not exclude the presence of metastases due to the low sensitivity of the test for the lateral compartment (56%) and medial compartment (6%). An undoubted advantage of US is the high specificity of 97% [45].

Traditional CT is used mainly in the search for MTC metastases in the lungs, mediastinal lymph nodes and liver [25,46]. CT does not play a major role in the diagnosis of primary tumors, showing both lower sensitivity (61.6% vs. 75.3%) and specificity (82.8% vs. 93.1%) than the US in the hands of an experienced diagnostician [43].

Magnetic resonance techniques are also not among the standard diagnostic options for MTC, as the available clinical studies from recent years are limited on this topic [47]. Instead, MRI is an excellent tool for the evaluation of liver and bone metastases [8,25].

### 1.10. Nuclear Medicine

The utility of nuclear medicine techniques in the diagnosis of MTC is not clearly defined. The ATA 2015 guidelines indicate that neither FDG-PET/CT (fluorodeoxyglucose–positron emission tomography) nor F-DOPA-PET/CT (4-dihydroxy-6-18F-fluoro-L-phenylalanine) are recommended for the detection of distant metastases, as these are less sensitive tests than other tests based on 2007 studies. In contrast, the ESMO 2019 guidelines classify F-DOPA-PET as a recommended test (when available) for the diagnosis of secondary foci [8,25]. In recent years, new clinical trials have been published that signal the potential benefit of using new agents from the field of nuclear medicine in the diagnosis of MTC, which were not included by the ATA 2015 and ESMO 2019 guidelines or were considered less relevant according to the state of the art at the time [48,49,50,51] (e.g., 68Ga-DOTA-MGS5-PET/CT [49], TF2/68Ga-IMP288-PET [50,51]).

18F-FDOPA-PET/CT is a compound similar in structure to naturally occurring amino acids. This compound is not specific to MTC, but this cancer shows high levels of the elements involved in its circulation—L-type amino acid transporter (LAT) and aromatic L-amino acid decarboxylase (AADC). This results in increased uptake of the tracer 18F-FDOPA into cancer cells and signal enhancement in PET/CT images [52,53]. 18F-FDOPA can be used to image secondary foci of MTC, particularly its recurrence. An extensive meta-analysis of the available data from 2020 shows a very high specificity but a very variable sensitivity, ranging between 45% and 93% depending on the study, which was probably due to the different technical aspects of the studies performed and the inclusion criteria [52]. It is noteworthy that sensitivity increases significantly (to approximately 90%) in patients with high serum calcitonin levels (>150 pq/mL) and in patients who had a doubling of calcitonin levels in less than 24 months [47,52]. 18F-FDOPA is also indicated as the single best modality for whole-body MTC metastasis detection [47] and as the best radiopharmaceutical PET among the five most commonly studied in the context of MTC [54]. The likely causes of false-negative results are MTC foci that are too small and foci located too close to natural tissues characterized by increased 18F-FDOPA content (striatum, liver, gallbladder, biliary tract, pancreas, kidneys, bowel, urinary tract). Other neuroendocrine tumors (NETs) that will also capture the tracer may be responsible for false-positive results [52].

Both 18F-FDG and 68Ga-SSA (somatostatin analog) are indicated as radiopharmaceuticals, which are less effective in combination with PET/CT than 18F-FDOPA, particularly because of their low sensitivity. Their use is limited to a select group of patients with secondary foci of MTC. 18F-FDG-PET/CT in particular may be useful when standard imaging modalities have failed, and a rapid rise in serum calcitonin and CEA is observed—doubling time < 1 year [52,54,55]. 68Ga-SSA, and in particular 68Ga-DOTA-TATE (DOTA-Tyr3-octreotate), is applicable in the diagnosis of MTC bone metastases. Here, DOTA-Tyr3-octreotate has shown better results than 18F-FDOPA and similar results to MRI [56,57,58].

A new potentially beneficial imaging modality in MTC is the use of minigastrin analogs targeting cholecystokinin-2 receptor (CCK2R) [47]. One of these analogs is 68Ga-DOTA-MGS5-PET/CT. The new DOTA-MGS5 was derived from the previously known DOTA-MG11. Until now, this compound has been mainly studied using animals, but single case reports about its use in humans with MTC with promising results are already available [59,60]. One of them reports the case of a patient with advanced MTC in whom 18F-FDOPA-PET/CT and 68Ga-DOTA-MGS5-PET/CT imaging modalities were used and compared. The extrahepatic lesions were better visualized with 18F-FDOPA, but 68Ga-DOTA-MGS5 visualized the hepatic lesions much more accurately. Furthermore, 68Ga-DOTA-MGS5 visualized two additional metastatic foci that were unremarkable with 18F-FDOPA. The physiological uptake of the 68Ga-DOTA-MGS5 tracer by the liver was low, resulting in images with very good contrast, especially two hours after radiopharmaceutical administration (compared to measurement after the first hour) [49].

Another developing branch of nuclear medicine in the diagnosis of MTC is pretargeted immuno-PET, which takes advantage of CEA secretion by MTC foci. Radiopharmaceuticals consist of murine or chimeric anti-CEA bispecific antibodies and pretargeted peptides connected with radioisotopes. The results of published studies are very promising. One study analyzed the TF2/68Ga-IMP288 combination. The overall sensitivity in diagnosing secondary MTC foci was estimated at 89%—for liver and lymph nodes 100%, for bone 87% and for lungs 42%. In the same study, only bone MRI achieved better overall sensitivity—CT 77%, bone MRI 92%, liver MRI 76% and 18F-FDOPA-PET/CT 66% [50]. Another study also using TF2/68Ga-IMP288 presented similar results. The overall sensitivity was 92% for immuno-PET, while for 18F-DOPA-PET/CT, it was 65%. Immuno-PET was found to be more sensitive than CT and 18F-FDOPA-PET/CT for localizing MTCs in lymph nodes (98% vs. 82% vs. 72%, respectively) and liver (98% vs. 87% vs. 65%, respectively) and more sensitive than MRI or 18F-FDOPA-PET/CT for localizing MTCs in bone (92% vs. 89% vs. 68%, respectively). However, the sensitivity of immuno-PET for lungs was much lower than that of CT (29% vs. 100%) [51]. TF2/68Ga-IMP288-PET/CT was also useful in confirming the metastatic nature of cardiac tumors in two other patients when 18F-FDOPA-PET/CT and 111In-octreoscan were inconclusive [61].

## 2. Surgical Treatment

### 2.1. Thyroidectomy

Surgical treatment is currently the first-choice method for the treatment of medullary thyroid carcinoma [5,25,33,62,63,64]. The primary technique cited by most of the papers and guidelines we found as the valid technique for initial treatment of MTC was total thyroidectomy (TT) combined with central compartment lymph node dissection (CCLND) [25,33,64,65,66,67,68]. The reason for choosing TT over hemithyroidectomy is the fact that multifocal and bilateral disease is present in most hereditary and 6% of sporadic MTC cases [5,62], and according to some sources, even in all hereditary and as many as 30% of sporadic MTC cases [7]. A slightly different approach was presented by a former Japanese researcher Miyauchi et al. [69], who suggested that, among patients with sporadic MTC who lack mutations in the RET gene, the chance of bilateral primary lesions is close to 0%. For this reason, the authors suggested hemithyroidectomy with systematic central and ipsilateral lymph node dissection instead of TT with CCLND [69]. This idea is currently not supported by most recent guidelines [8,25,33,68], presumably because of the aforementioned reason. Interestingly, however, the Japan Association of Endocrine Surgeons still recommends in their latest guideline (2020) to perform lobectomy instead of TT among patients with sporadic MTC located in only one lobe, citing the study of the aforementioned Miyauchi et al. and the lack of studies showing the superiority of TT over lobectomy for sporadic MTC [70]. New studies comparing the two methods should be performed to unify international guidelines.

TT may also be indicated as prophylactic treatment in patients diagnosed with a mutation in the RET gene. In children, depending on the RET mutation category, TT is recommended depending on age and taking other factors into account. Then, a stand-alone TT can be performed without CCLND. In adults with RET mutations, TT and the excision of the appropriate lymph nodes is recommended depending on Ctn levels [25].

TT is also recommended for patients who initially undergo lobectomy and are subsequently found postoperatively to have an RET germline mutation, elevated serum Ctn level or imaging study results suggesting residual MTC [25].

### 2.2. Central Compartment Lymph Node Dissection

TT combined with CCLND (level VI) is recommended for any MTC by ATA 2015 and the United Kingdom Multidisciplinary Guidelines 2016 [25,68]. Recommendations from the ATA are, at the same time, among the most frequently described recommendations by contemporary works as a standard of practice [64,65,66,67]. This method is justified by the presence of central lymph node metastases in 50–70% of both patients with sporadic MTC and those with hereditary MTC, regardless of whether the primary tumor is smaller than 1 cm or larger than 4 cm [62]. However, there is now a widespread trend in recent recommendations to discontinue CCLND and use only TT when preoperative Ctn levels are <20 pg/mL [5,8,33,62]. This trend is based on work that indicated that there is no risk of central lymph node metastasis when Ctn levels are <20 pg/mL [71].

### 2.3. Lateral Compartment Lymph Node Dissection

Currently, lateral compartment lymph node dissection (LCLND) is the most controversial [5,7,25,33,64,65,72,73]. One view on deciding whether to perform prophylactic LCLND when there is no evidence of neck metastases on the US is based on the Ctn level [5,8,25,70]. The ATA and the Japan Association of Endocrine Surgeons do not place specific limits on whether LCLND should be performed but only recommend relying on Ctn level values to make this decision [25,70]. In contrast, the ESMO (Table 4 [8]) recommends specific procedures for deciding on whether to perform LCLND depending on Ctn level cutoffs. In the presence of cervical lymph node metastases, the ATA and ESMO recommend TT combined with CCLND and the excision of the involved lateral lymph nodes. In cases of the involvement of the ipsilateral lymph nodes but not the contralateral lymph nodes, the excision of the contralateral compartment at Ctn > 200 pg/mL is recommended [8,25].

In addition to Ctn, many other investigators have offered alternative surgical regimens for MTC and the surrounding lymph nodes [7,33]. Citing the 2021 NCCN guideline, Kim et al. [33] in their paper suggested TT and the consideration of CCLND for unilateral MTCs less than 1 cm. For MTCs that are bilateral or ≥1 cm, TT + bilateral CCLND is recommended. However, for patients with MEN 2a and familial MTC with tumors > 1 cm or central lymph node metastases, TT + CCLND + LCLND is recommended [33]. However, the incidence of central compartment lymph node metastases is estimated to be as high as 50–75%, regardless of whether the tumor size is <1 cm or >4 cm [62]. Other papers and guidelines rely on tumor size, blood Ctn level, lymph node involvement and the T classification of the tumor in various combinations to make decisions about whether to perform LCLND [7,68,70]. A completely different approach was recently presented by Niederle et al. [63] in their 2021 paper. They recommended using a desmoplastic stromal reaction (DSR) assessment of the tumor to decide on the extent of surgery. Based on their study of 360 patients with MTC, they noted that patients who were classified intraoperatively as DSR negative did not have neck lymph node metastases or distant metastases. In contrast, 31.4% of DSR-positive patients had lymph node metastases, and 6.4% had distant metastases (*p* < 0.001 and *p* = 0.031 relative to DSR-negative patients) [63]. Similarly, the use of DSR as a possible predictor for lateral lymph node metastases was suggested by Opsahl et al. [74] in their 2019 paper. Furthermore, in their paper, they argued that Ctn is not an optimal biomarker to assess the need for LCLND. They suggest, as a reason, that there is no specific cutoff from which we can confidently approve or advise against LCLND [74].

However, there are several papers that do not support prophylactic LCLND [64,65,72]. Yamashita et al. [64] in their study showed that prophylactic LCLND in the absence of structural disease had no effect on overall recurrence, locoregional recurrence or even overall survival in both patients with sporadic MTC and those with hereditary MTC relative to patients who did not undergo this surgery. Moreover, the majority of patients in the group that did not undergo LCLND met the criteria (according to ATA) for the group where LCLND should be considered. The development of cancer requiring later LCLND in the group not subjected to prophylactic LCLND was observed in only a small percentage of patients but without any effect on overall survival [64]. Similarly, Spanheimer’s et al. [72] in their study found no relationship between prophylactic LCLND and better patient outcomes. This study compared 89 patients who initially had Ctn levels > 200 pg/mL. A total of 45 of these patients underwent LCLND and 44 did not. There was no statistically significant difference between groups in the 10-year incidence of recurrence in the neck (20.9% LCLND vs. 30.4%, *p* = 0.46), incidence of distant recurrence (18.3% vs. 18.4%, *p* = 0.97) or overall survival (82% vs. 93%, *p* = 0.6). Interestingly, it was shown that the incidence of recurrence in the neck was at a similar level in both groups for the first 5 years and then increased among the group who did not undergo prophylactic LCLND. These results could suggest that LCLND leads to the excision of microscopic disease, which alone does not significantly increase the risk of distant metastases [72]. Pena et al. [65] in their study on 66 patients with sporadic MTC, of whom 44 were in the observation group and 22 were in the LCLND group, also showed no effect of LCLND on biochemical cure (observation group vs. LCLND group: 82% vs. 85%, *p* > 0.999), locoregional recurrence (5% vs. 5%, *p* > 0.999), distant metastasis (9% vs. 5%, *p* > 0.999) or 10-year overall survival (0.84 vs. 0.93 *p* = 0.156). Moreover, from the observation group, 93% of patients showed a preoperative Ctn level > 20 pg/mL, and 68% showed a preoperative Ctn level > 200 pg/mL. In the prophylactic LCLND group, 95% and 95% of patients had preoperative Ctn levels > 20 pg/mL and >200 pg/mL, respectively. Moreover, performing prophylactic LCLND is associated with increased operative time, cost and postoperative complications [65]. Van Beek et al. [66] in their study found that performing TT with LCLND was associated with a higher risk of at least transient hypoparathyroidism (31.1% vs. 19%) and recurrent laryngeal nerve (RLN) palsy (14.2% vs. 2%) than performing TT without LCLND. In addition, patients who underwent TT + CCLND + LCLND were more likely to develop at least transient RLN palsy (21.2% vs. 8.1%) than patients who underwent TT + CCLND alone. However, there was no difference in the rate of developing at least transient hypoparathyroidism between the two groups [66].

### 2.4. Distant Metastasis

Primary tumor resection in patients with unresectable distant metastases also remains controversial [75]. In this case, the 2015 Revised ATA Guidelines recommended less aggressive surgery in the central and lateral lymph node compartments to avoid adverse side effects [25]. However, Zhuang et al. [75] in their paper found that primary tumor resection in these patients not only has a palliative role but also prolongs their median overall survival, unlike most palliative surgeries. Unfortunately, they did not specify exactly which lymph node compartments were resected in his patients [75]. Thus, for such patients, the benefits of prolonging their lives would have to be weighed individually, taking into account the increased risk of postoperative complications and their quality of life.

### 2.5. Transoral Surgery

Transoral endoscopic thyroidectomy vestibular approach (TOETVA) is a widely recognized and recently highly developed alternative to open thyroidectomy. The indications for this method are primarily the patient’s desire to avoid scarring and primary papillary microcarcinoma without local or distant metastasis. Both TT and CCLND can be performed with this method. However, this method has many contraindications, including distant or lateral neck metastases [76], as well as MTC [77], among others. Nonetheless, the possibility that MTC will also be treated with this method in the future cannot be excluded. Chen et al. [78] in their paper reported the case of a 33-year-old female patient who was found to have a lobe thyroid nodule on US examination in the absence of surrounding lymph node and tissue involvement. She underwent unilateral lobectomy and isthmectomy by the TOETVA. However, due to an intraoperative suspicion of MTC, TT and CCLND were chosen. Postoperatively, the patient was diagnosed with MTC pT1N0M0, size 1.6 cm in diameter. The day after the patient’s surgery, her Ctn level dropped from a preoperative level of 409.5 pg/mL to 16.4 pg/mL. The only postoperative complication noted was transient hypoparathyroidism. At the 6-month follow-up, no recurrence was noted. The researchers concluded that the TOETVA could be used for treatment in patients with cT1N0M0 MTC and high cosmetic need, as for patients with cT1N0M0 MTC, the treatment modality is TT + CCLND, which can be performed by TOETVA [78]. The use of transoral robotic thyroidectomy may also soon find application for MTC surgery, as surgeons’ experience with this method increases [79]. This method would be a great convenience for patients, as scarless surgery has a high cosmetic satisfaction among patients and is readily chosen, especially by young women [80].

### 2.6. Intraoperative Parathyroid Gland Identification

Temporary hypoparathyroidism is one of the most common complications after thyroidectomy. Temporary hypoparathyroidism can occur in 20–30% of patients, and hypoparathyroidism can become permanent in 1–4% of patients [81]. During TT, surgeons try to preserve healthy parathyroid glands (PGs). For this purpose, healthy PGs are currently mostly identified based on the surgeon’s experience and visual differences between PGs and the surrounding tissue [82,83,84]. Recently, there has been significant development in the techniques for the intraoperative visualization of PGs. Currently, a very popular technique is the use of fluorescence with near-infrared light. For this method, surgeons can use both autofluorescence and contrast-enhanced fluorescence [82]. For contrast-enhanced fluorescence, indocyanine green angiography (ICGA) is mainly used. There are many studies on the efficacy of intraoperative imaging of PGs using this method during thyroid or parathyroid surgery, where PG detection rates ranged from 31/71 to 100% [83]. For example, Rudin et al. [85] studied 210 patients who underwent total/near-total thyroidectomy. In 86 of these patients, ICGA was performed intraoperatively to identify PGs. Using ICGA, 281 of 344 PGs (82%) were detected. A total of 36% of patients in the control group developed postoperative biochemical hypoparathyroidism (PTH < 15 pg/mL), and 10% showed a postoperative PTH < 6 pg/mL. In the ICGA group, 37% and 15% of patients developed biochemical hypoparathyroidism (PTH < 15 pg/mL) and a postoperative PTH < 6 pg/mL, respectively. Persistent hypoparathyroidism developed in one patient from each group [85].

Another technique to visualize PGs is the use of near-infrared autofluorescence (NIRAF) [86,87,88,89]. In their study using a fiber optic probe based on tissues collected from 110 patients, McWade et al. [90] demonstrated that PGs show a signal 1.2–25 times higher than the surrounding tissues. The sensitivity was 100% [90]. A study is currently underway on the utility of this method for intraoperative PG detection during TT (NCT04281875). Another method of using NIRAF to visualize PGs is by using a near-infrared camera. Benmiloud et al. [91] studied 245 patients during TT with or without lymph node dissection, and this method was used in 121 of the patients. Postoperatively, they found that hypocalcemia was significantly lower in patients in the NIRAF group than in those in the standard-care group (9.1% vs. 21.7%, *p* = 0.007). All four PGs were also detected more frequently in patients in the NIRAF group (47.1% vs. 19.2%, *p* < 0.001). However, there was no significant difference in PTH concentration between the groups on the first day after surgery [91]. These methods, which have recently been highly developed, offer the opportunity for better intraoperative visualization of PGs and thus reduce the proportion of patients who will develop hypoparathyroidism after TT performed for MTC.

## 3. Systemic Treatment

Targeted therapies are a management option for patients with progressive or symptomatic disease with locoregional or metastatic MTC [33]. Targeted therapies have helped supplant cytotoxic chemotherapy, which has low efficacy against disease progression, but there is still the problem of toxic side effects and frequent development of tumor resistance when using the current treatment regimen [92]. These problems indicate that there is still a need to develop an effective treatment for MTC. Below, we summarize the available systemic therapeutic options and current ongoing clinical trials with a focus on multikinase inhibitors (MKIs), highly selective RET inhibitors, radionuclide therapy and immunotherapy.

### 3.1. Multikinase Inhibitors

Mutations in the RET protooncogene lead to the overexpression of the receptor tyrosine kinase, resulting in increased activity in the cellular pathways responsible for proliferation, angiogenesis and apoptosis [93]. This process is responsible for tumorigenesis in all described cases of hereditary MTC and 40–50% of sporadic MTC cases [94].

Consistent with this knowledge, multikinase inhibitor trials were conducted that resulted in Food and Drug Administration (FDA) approval of two systemic therapies, cabozantinib and vandetanib, for the treatment of progressive or symptomatic MTC with locally advanced or metastatic disease [95,96,97]. The results from the EXAM (thyroid cancer) trial and the ZETA trial showed improved disease progression-free survival (PFS) compared to placebo (*p* < 0.001); unfortunately, there was no significant survival benefit for either drug compared to placebo [98]. However, in both phase III trials evaluating vandetanib and cabozantinib, overall survival (OS) was a secondary end point. In patients with advanced and progressing neoplastic disease, one of the main goals is to achieve stable disease [95,96]. Vandetanib inhibits the activity of RET, as well as other receptor tyrosine kinases, including vascular endothelial growth factor receptors (VEGFR-2, VEGFR-3) and epidermal growth factor receptor (EGFR). Because of this broad kinase inhibitor activity, there are limitations to this therapy, which include intolerable side effects and the development of resistance to treatment. Cabozantinib has been found to be effective in patients with the RETM918T mutation, which is one of the causes of resistance to multidrug therapy [99].

Studies of new MKIs are ongoing [100,101,102]. A phase II study evaluating the safety and efficacy of lenvatinib demonstrated a median PFS of 9 months in patients with unresectable MTC. The study included patients with both RET- and RAS-driven disease. There was no significant improvement in the overall survival (OS) [103]. The phase III trial of this study is now ongoing (NCT00784303). Recently, a study of lenvatinib showed interesting results as a salvage therapy in patients with advanced MTC who lost clinical benefit with other TKIs [104]. A multicenter, randomized phase IIIB trial (ALTER 01031 and NCT02586350) of anlotinib showed interesting results. Anlotinib demonstrated significantly prolonged PFS in comparison to the placebo (20.7 months vs. 11.1 months; HR, 0.53; 95% CI, 0.30–0.95). The ORR in anlotinib group was 48.4%. The most common adverse events were palmar–plantar erythrodysesthesia syndrome (62.9%), proteinuria (61.3%) and hypertriglyceridemia (48.4%). The mutation status in this study has not been determined [105].

Multitarget drugs are currently being evaluated in several interesting and promising studies. Regorafenib is being evaluated in a phase II trial as a second- or third-line therapy in metastatic MTC. The FDA has approved regorafenib for the treatment of metastatic colorectal cancer and locally advanced, unresectable or metastatic gastrointestinal lining tumors. Due to its multiple kinase inhibitory properties, an evaluation of regorafenib as a potential treatment for thyroid cancer is warranted (NCT02657551).

Immunotherapeutic approaches, which are discussed at length in the following sections, may also be an important treatment option for RET mutations. The inhibition of the mortalin molecule results in the suppression of medullary thyroid cancer cells, inducing apoptosis and the downregulation of RET. Due to its similar effects, the agent MKT-077 may be used as a molecular therapy in MTC. Unfortunately, this molecule is toxic to animals. Analogs of MKT-077 (JG-98 and JG-194) were recently tested, and the results were promising. Both inhibited tumor cell proliferation in vandetanib- and cabozantinib-resistant MTC [106].

As mentioned above, the greatest obstacle appears to be the development of resistance during treatment with multikinase therapy. Researchers suggested that the use of more specific drugs could provide therapeutic options for patients with metastatic, progressive MTC.

### 3.2. Highly Selective RET Inhibitors

Two specific RET kinase inhibitors, selpercatinib (LOXO-292) and pralsetinib (BLU-667), were approved by the FDA for the treatment of RET-mutated MTC in 2020 [107,108]. Selpercatinib was studied in the multicenter phase I/II clinical trial LIBRETTO-001 and showed favorable results compared to previous multikinase inhibitors. Patients with RET-mutant medullary thyroid cancer who had previously received vandetanib, cabozantinib or both achieved an overall response rate (ORR) of 69% (95% confidence interval (CI), 55 to 81) and a 1-year PFS of 82% (95% CI, 69 to 90). Patients with RET-mutated medullary thyroid cancer who had not previously received vandetanib or cabozantinib had an ORR of 73% (95% CI, 62 to 82) and a PFS of 92% (95% CI, 82 to 97) [109]. The registrational phase I/II ARROW trial, with the use of pralsetinib, in patients with MTC demonstrated rapid, potent and durable clinical activity, regardless of RET mutation (NCT03037385) [110]. These new RET-specific inhibitors have thus far shown a better side effect profile, which is probably due to their high selectivity with VEGFR bypass [92]. Previous generation of MKIs were not selective and were also blocking the VEGFR factor, part of the angiogenic and proliferative pathways. This caused intolerable sides effects, such as diarrhea, proteinuria, fatigue and hypertension [92]. Promisingly, when using selpercatinib in the LIBRETTO-001 study, researchers demonstrated a treatment-related adverse event that caused an overall drop-out rate in 2% of patients [109]. Unfortunately, disease that is resistant to the new TKIs has also developed. According to one study, RET G810 mutation in patients with RET fusion-positive non-small-cell lung carcinoma (NSCLC) and RET-mutation-positive MTC leads to acquiring recurrent mechanism of resistance to selpercatinib [111]. Based on a different study, gradual, multifactorial acquisition of resistance is more reasonable than the currently predominant interpretation of resistance mechanism based on a point mutation. Researchers think that the cooperation of multigenetic and epigenetic changes plays a bigger role in the mechanism of developing resistance to TKIs. They found that resistance can originate from heterogenous subpopulations with variable TKI’s sensitivity tumor tissue. Additionally, during the evolution of resistance, tumor cells present unique features that can be temporary treatment opportunity [112]. This preclinical study was modeled using ALK-positive NSCLC patient-derived xenograft, but thanks to this insightful analysis, it is possible to incorporate this knowledge for the treatment of MTC. Without a doubt, there is a need to expand our understanding of resistance to targeted therapies.

Based on the above information, new clinical trials are being conducted in malignant cancer patients with RET mutations. TPX-0046 has dual inhibitory activity against RET and SRC, including the RET G810 mutation. A phase I/II, first-in-human, open-label study evaluating the safety and efficacy of TPX-0046 is currently recruiting patients (NCT04161391).

TAS0953/HM06 is another selective RET inhibitor that is being studied in a phase I/II trial (MARGARET) in patients with advanced solid tumors with RET gene mutations (NCT04683250).

BOS-172738 also inhibits the RET gene. The results from a phase I trial demonstrated potent antitumor activity and a well-tolerated safety profile in the treatment of RET gene-altered MTC and NSCLC [113]. Of the 30 patients with MTC who were evaluated, the ORR was 44% in 16 patients (NCT03780517).

Regarding the MET mutation causing resistance, according to a study, the combination of selpercatinib and MET inhibitor crizotinib was able to overcome drug resistance [114].

### 3.3. Targeting RAS-Mutated MTC

To circumvent resistance mechanisms, researchers have investigated another mutation that leads to resistance to kinase inhibitors. RAS mutations, mainly HRAS and KRAS, lead to approximately 40% of sporadic MTC cases without RET mutations [115]. The remaining 20% of cases have no identified oncogenic agent, meaning that they cannot be eradicated by blocking the aforementioned factors, and new therapies still need to be found [116]. RAS genes are responsible for the production of proteins that control cell signaling pathways [117]. The overexpression of their products leads to tumor formation and is associated with the appearance of MTC [118]. A major problem in RAS-specific therapy is that these molecules are difficult to target pharmacologically [119]. A key issue for the efficacy of highly selective RAS inhibitors is to find a molecule that overcomes this molecular inaccessibility. The underlying reason found in recent studies was that HRAS, unlike KRAS, is prenylated only by farnesyltransferase [119]. Tipifarnib, a farnesyltransferase inhibitor, was tested in an open-label phase II trial in patients with HRAS-driven cancers and in patients with HRAS-mutant MTC. This study has been completed, and the results are now being published (NCT02383927).

### 3.4. Immunotherapy

Immunotherapy approaches are now underway for the treatment of patients with MTC. The above-mentioned mutations of RAS have also been targeted with the use of these methods. In a preliminary study, a polyclonal CD8+ T cell was identified to act against mutant KRAS G12D in tumor-infiltrating lymphocytes obtained from a patient with metastatic colorectal cancer [120]. Based on this trial, other early stage trials are now underway. In a phase I/II study, the researchers administered peripheral blood lymphocytes transduced with a murine T-cell receptor recognizing the G12V variant of mutated KRAS (NCT03190941).

Another immunological approach was focused on the identification of tumor-specific antigens that could be blocked with tumor vaccines. One of them was investigated in an early phase study that incorporated calcitonin and CEA, MTC secretory products. The vaccine was injected into seven patients. Calcitonin and CEA were markedly decreased in three out of seven patients, with one patient showing a complete regression of metastatic MTC [121]. Novel approaches that use this method, including a biological recombinant *Saccharomyces cerevisiae*-CEA vaccine (GI-6207), are being tested. One trial is active but not yet recruiting (NCT01856920).

Bhoj et al. [122] used the chimeric antigen T-cell receptor (CAR-T) immunotherapy. The results were encouraging, and the authors reported that CAR T targeting glial-derived neurotrophic factor (GDNF) family receptor alpha 4 (GFRα4) could eliminate MTC in a murine xenograft model [122]. An open-label phase I study with CAR T-GFRa4 cells in metastatic CAR T-GFRa4 cells is now recruiting (NCT04877613).

Inhibiting the immune checkpoints is also taken into consideration as an MTC immunotherapy model. Programed cell death protein-1 (PD-1) and cytotoxic T-lymphocyte antigen 4 (CTLA-4) are responsible for inhibitory cell-cycle pathways [123]. Blocking these inhibitory pathways enhances effector T cells and inhibits regulatory suppressor cells [124]. Their blockade is a promising immunotherapeutic method that is used in other cancer types, but in medullary thyroid carcinoma, the studies supporting this method are insufficient [125]. In a recently conducted study, 200 patients who received surgery before systemic treatment were tested. The results showed that PD-1 positivity was detected in 27 (13.5%) patients, and cytotoxic T-lymphocyte antigen 4 (CTLA-4) positivity was detected in 25 (12.5%) patients, with expression being positively correlated with positive detection [126]. This identification is very promising in terms of tumor immunotherapy. The antibody anti-PD-1 pembrolizumab (MK-3475) has been recently tested as a monotherapy in advanced solid tumor patients, including a group of thyroid cancer patients. The results were submitted in 2022 and are now waiting for a quality control review (NCT02054806).

### 3.5. External Beam Radiation Therapy

External beam radiation therapy (EBRT) is a well-known treatment for MTC foci that is included in the ATA 2015 guidelines. It is recommended to consider postoperative adjuvant EBRT on the neck and mediastinum in a select group of patients who are at high risk of local recurrence of MTC and have not undergone complete primary MTC resection but are at risk of upper airway obstruction. Indications for EBRT further include brain, bone, skin metastases and therapy in palliative patients [25]. The use of ERBT should be considered individually for each patient, weighing the potential benefits and losses. There are studies verifying the use of postoperative EBRT in selected patients, with some results indicating benefits and some not recognizing benefits [127,128,129].

### 3.6. Nuclear Medicine

Pretargeted radioimmunotherapy appears to be a promising new therapeutic modality [130]. The ATA 2015 and ESMO 2019 guidelines recommend its consideration in selected patients [8,25]. For a long time, the peptide receptor radionuclide therapy (PRRT) mainly used radionuclides (90Y, 177Lu) conjugated with somatostatin analogs (SSAs). Studies involving these compounds in the treatment of MTC are limited, but the results of the available studies show a significant disease control rate in 62.4% of patients. With the discovery of CCK2Rs, there is a new opportunity to target PRRT at these receptors. Most studies to date have been conducted using animals. Despite promising results, the considerable nephrotoxicity of the therapy remains a major problem, although studies have also reported low patient harm [131,132,133].

One of the compounds studied is 177Lu-DOTA-TATE. In a group of 43 patients, the response of MTC foci was evaluated according to symptoms, serum markers and imaging in 68Ga-DOTA-TATE-PET/CT and 18F-FDG-PET/CT. At least a partial improvement in symptoms was observed in 47% of subjects, and improvement in biochemical parameters was observed in 41%. In terms of both parameters, deterioration occurred in 49%. In the remainder, the disease remained stable. In terms of objective imaging, deterioration occurred in 39% of patients, 10% had improvement, and 51% had no significant change. It should be noted that, in this study, a significantly better response was observed in patients with a calcitonin doubling time > 24 months than in those with a doubling time < 24 months (median progression-free survival not yet reached vs. 10 months and median overall survival 60 months vs. 20 months, respectively) [133]. Another study on seven patients using the same radiopharmaceutical showed disease progression in only one patient. In the others, the disease course stabilized. A molecular response according to the European Organization for Research and Treatment of Cancer (EORTC) criteria was observed in all seven patients. A biochemical response occurred in five patients. Patients mostly tolerated the therapy well [134].

A phase 0 clinical trial conducted on six patients using the minigastrin analog 177Lu-PP-F11N was published in 2020. The trial demonstrated that radiotherapeutic agents accumulated in MTC foci in amounts theoretically sufficient for effective therapy. The radiotherapeutic agent was detected by in vitro autoradiography even in lesions too small and thus invisible in 177Lu-PP-F11N-SPECT (single-photon emission computed tomography). The low acute toxicity and the low harm of the radiopharmaceutical to the kidney and bone marrow proved to be a great advantage. The organ with the highest sensitivity appeared to be the stomach [135]. In the future, it may be possible to further enhance the efficacy of this drug by combining it with inhibitors of mammalian target of rapamycin in complex 1 (mTORC1), which was found to increase the drug uptake of cancer cells in animal models using cancer cells with CCK2R [136].

## 4. Conclusions

Despite the recently increased interest in other types of thyroid cancer, there are still many advances in the diagnosis and treatment of MTC. The development of modern surgical techniques and tumor imaging techniques and the use of new compounds in nuclear medicine or in treatment are advances bringing much to the management of MTC. Although there are many guidelines and standards for dealing with MTC, it is still a dynamic field and one in which much has changed recently. Further research into the development of new treatments and diagnostics, as well as the standardization of MTC management, is desirable.

## Figures and Tables

**Table 1 cancers-14-03643-t001:** UICC TNM classification.

T—Primary Tumor	T/N/M	Characterization
	T1	Tumor 2 cm or less in greatest dimension, limited to the thyroid
	T1a	Tumor ≤ 1 cm in greatest dimension, limited to the thyroid
	T1b	Tumor > 1 cm but ≤2 cm in greatest dimension, limited to the thyroid
	T2	Tumor > 2 cm but ≤4 cm in greatest dimension, limited to the thyroid
	T3	Tumor > 4 cm in greatest dimension, limited to the thyroid or with gross extrathyroidal extension invading only strap muscles (sternohyoid, sternothyroid or omohyoid muscles)
	T4a	Tumor extends beyond the thyroid capsule and invades any of the following: subcutaneous soft tissues, larynx, trachea, esophagus, recurrent laryngeal nerve
	T4b	Tumor invades prevertebral fascia or encasing the carotid artery or mediastinal vessels from a tumor of any size
N—regional lymph nodes		
	N0	No evidence of locoregional lymph node metastasis
	N1a	Metastasis to level VI (pretracheal, paratracheal and prelaryngeal/Delphian lymph nodes) or upper/superior mediastinum
	N1b	Metastasis in other unilateral, bilateral or contralateral cervical compartments (levels I, II, III, IV or V) or retropharyngeal
M—distant metastasis		
	M0	No distant metastasis
	M1	Distant metastasis

**Table 2 cancers-14-03643-t002:** MTC staging system.

Stage	T (Primary Tumor)	N (Regional Lymph Nodes)	M (Distant Metastasis)
I	T1a, T1b	N0	M0
II	T2, T3	N0	M0
III	T1–T3	N1a	M0
IVA	T1–T3	N1b	M0
	T4	Any N	M0
IVB	T4b	Any N	M0
IVC	Any T	Any N	M1

**Table 3 cancers-14-03643-t003:** Comparison of the sensitivity of MTC detection.

Diagnostics	Sensitivity		Annotations
US	Primary tumor	75–90%	Standard procedure
	Lateral neck LN	56%	
	Central neck LN	6%	
US + serum Ctn and CEA	Primary tumor	95%	
CT	Overall	77–85%	Standard procedure
	LN	82%	
	Liver	87%	
	Bones	-	
	Lungs	100%	
MRI	Bones	89–92%	Standard procedure
	Liver	76–89%	
18F-FDOPA-PET/CT	Overall	45–93%	ATA 2015: not recommended ESMO 2019: recommended
	LN	72%	
	Liver	65%	
	Bones	68%	
	Lungs	14%	
	Lateral neck LN	75%	
	Central neck LN	28%	
68Ga-DOTA-TATE-PET/CT	Overall	84%	New
	Neck LN	56–63%	
	Mediastinal LN	100%	
	Liver	9%	
	Bones	100%	
	Lungs	57–63%	
68Ga-DOTA-MGS5-PET/CT	Not enough data		New
68Ga-IMP288-PET/CT	Overall	89–92%	New
	LN	98–100%	
	Liver	98–100%	
	Bones	87–92%	
	Lungs	29–42%	

MTC: medullary thyroid cancer; CEA: carcinoembryonic antigen; Ctn: calcitonin; CT: computed tomography; LN: lymph nodes; MRI: magnetic resonance imaging; PET: positron emission tomography.

**Table 4 cancers-14-03643-t004:** Procedure for the treatment of medullary thyroid carcinoma based on the calcitonin level, as recommended by ESMO.

Calcitonin Level [pg/mL]	Procedure for MTC Treatment		
	Neck US—Negative		Neck US—Positive
<20	TT		TT + bilateral CCLND + dissection of involved levels
20–50	TT +/− bilateral CCLND		
50–200	TT + bilateral CCLND + ipsilateral LCLND *		
200–500	TT + bilateral CCLND + bilateral LCLND *		TT + bilateral CCLND + dissection of involved levels + contralateral lymph node dissection
>500	M0	M1	
	TT + bilateral CCLND + bilateral LCLND *	Range of surgery based on disease progression and symptoms	

* At least IIa, III, IV. TT: total thyroidectomy; CCLND: central compartment lymph node dissection; LCLND: lateral compartment lymph node dissection.

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
