# Peer review of "Update on the Diagnosis and Management of Medullary Thyroid Cancer: What Has Changed in Recent Years?"

_cancers, 2022, doi:10.3390/cancers14153643_

Round 1

Reviewer 1 Report

The review article by Kaliszewski et al. summarizes the current state of understanding on diagnosis and management of medullary thyroid cancer (MTC). The article is interesting, however, it requires some minor changes (listed below).

1. The Authors should pay more attention to references. For example, in lines 333 and 341, the Authors refers to the paper by Miyauchi, however, I cannot find this article in the References section. Similarly, I cannot find the paper written by Szabo mentioned in line 405. In line 539 it is worth citing an additional more recent paper, e.g. doi: 10.3390/ijms222111829. Moreover, it is worth adding a reference in line 129 (i.e., “…86% [Ref]”). I think Ref [90], i.e., Wirth et al., 2020, in line 568 should be moved to line 573. Whereas, I would suggest referring in line 568 to other article. Finally, I/II ARROW trial is incorrectly cited in line 578.

2. In line 384 the Authors states that "Kim in his paper suggested...", which suggests that the cited publication has only one author, which is not true. It is better to replace it with Kim et al. suggested...". A similar correction will be needed in other parts of the manuscript (e.g., in lines 393, 400, 413, 422, 432, 443, 458, 501).

3. I would suggest rephrasing lines 183-186.

4. I wonder why the Authors sometimes provide p value and sometimes not. Please explain/correct this.

5. Line 210: “miR12 and miR-29-5p” are incorrect. It should be replaced with “miR-127 and miR-129-5p”.

6. Lines 479-481: It seems to me that these two sentences should be combined into one.

7. Line 664: The meaning of “SSAs” should be explained.

Examples of typos/other minor errors:

1. Line 199: “8(PAX8)” should be changed for “8 (PAX8)”.

2. The Authors should standardize the notation of minority and majority signs and numbers. Sometimes there is a space between the sign and the number (e.g.,"< 1 year" in line 285) and sometimes not (e.g., "p>0.999” in line 426).

3. “ml” (e.g., in line 192) and “mL” (e.g., in line 375) are used interchangeably. Please standardise this.

4. “However” repeats in line 455 and 456. I would suggest changing it.

5. Lines 579 and 584: “RETG810” should be changed for “RET G810”.

6. Line 644: “in in 25” should be corrected.

Author Response

Journal: Cancers                                                                                 July 18, 2022

MDPI

Special Issue: “Advances in Thyroid Cancers”

Dear Editors: Prof. Stefania Masone, Prof. Nunzio Velotti

Dear Editors and Reviewers,

At the very beginning we would like to thank you very much for the possibility to re-submit our revised manuscript entitled Update on the Diagnosis and Management of Medullary Thyroid Cancer: What has Changed in Recent Years?” Thank you very much for considering it for potential publication in Cancers.

We would like to thank the reviewers for the very thorough reviews and for the advices and constructive criticism, which have been valuable for improving our paper. All of the suggestions for changes and improvements were very helpful to us, and we have revised the manuscript according to the recommendations made in the reviews. All of the changed, deleted and added portions of the manuscript are marked by using Track Changes. According to the reviewers’ instructions we corrected our manuscript point-by-point as follows.

Reviewer 1

Dear Reviewer, thank you very much for your review and suggestions. Thank you very much for the statement, that “the article is interesting”, it is wonderful and important opinion for us. Thank you.  As far as your suggestions, we introduced all of them to the manuscript as follows:

    1. The Authors should pay more attention to references. For example, in lines 333 and 341, the Authors refers to the paper by Miyauchi, however, I cannot find this article in the References section. Similarly, I cannot find the paper written by Szabo mentioned in line In line 539 it is worth citing an additional more recent paper, e.g. doi: 10.3390/ijms222111829. Moreover, it is worth adding a reference in line 129 (i.e., “…86% [Ref]”). I think Ref [90], i.e., Wirth et al., 2020, in line 568 should be moved to line 573. Whereas, I would suggest referring in line 568 to other article. Finally, I/II ARROW trial is incorrectly cited in line 578.
    2. Thank you for your comment. We did not include the paper by Miyauchi in the References, despite the fact of referring to it, because we cited the article, which was referring to the article of this author. However we have changed it, and referred to the original work in References as it have should been done before. Second mistake i.e. not referring to Szabo’s paper unfortunately was done, because of confusing the name with the surname of the author. We indeed referred to his work, but using his surname instead of his name. We have changed it. Thank you. Indeed, a reference in line 129 was needed. We fixed this mistake, and referred to paper doi: 10.1159/000515373. We also added next reference mentioned by you. We have removed Ref [90] (previous number), i.e., Wirth et al., 2020, from line you have pointed at. In this place we added new, more accurate articles as you suggested. Through our mistake the [90] (previous number) citation was misattributed, and we moved it to the accurate place in line, as you suggested. We also have more precisely cited I/II ARROW trial with regarding to the medullary thyroid cancer (MTC) treatment. Thank you for your valuable suggestions.
    1. In line 384 the Authors states that "Kim in his paper suggested...", which suggests that the cited publication has only one author, which is not true. It is better to replace it with Kim et al. suggested...". A similar correction will be needed in other parts of the manuscript (e.g., in lines 393, 400, 413, 422, 432, 443, 458, 501).
    2. Thank you for this comment. We corrected this mistake in the places mentioned by you and in the other places, where we have found it.
    1. I would suggest rephrasing lines 183-186.
    2. We rephrased mentioned text in order to make the paper more clear and more comfortable to read. Thank you very much for this suggestion.
    1. I wonder why the Authors sometimes provide p value and sometimes not. Please explain/correct this.
    2. Thank you for this comment. In some articles which we mentioned, we couldn’t find the p value. That is why sometimes there it was presented and sometimes not. Our article was written by more than one author, so some of us preferred to use the p value more frequently than others. However due to your suggestion, we have advised authors to search through the mentioned papers and include p value, if possible.
    1. Line 210: “miR12 and miR-29-5p” are incorrect. It should be replaced with “miR-127 and miR-129-5p”
    2. Thank you very much for noticing this mistake. We corrected the information to be compatible with the information included in the reference.
    1. Lines 479-481: It seems to me that these two sentences should be combined into one.
    2. Indeed, these two sentences contained similar information. We have deleted the second sentence. Thank you
    1. Line 664: The meaning of “SSAs” should be explained.
    2. We explained that “SSAs” as “somatostatin analogues” in the text. Thank you.

Minor errors:

    1. Line 199: “8(PAX8)” should be changed for “8 (PAX8)”.
    2. We fixed it as you suggested.
    1. The Authors should standardize the notation of minority and majority signs and numbers. Sometimes there is a space between the sign and the number (e.g.,"< 1 year" in line 285) and sometimes not (e.g., "p>999” in line 426).
    2. Thank you for spotting these inaccuracies. We have standardized all the differences.
    1. “ml” (e.g., in line 192) and “mL” (e.g., in line 375) are used interchangeably. Please standardise this.
    2. We have standardized this errors. Thank you.

    1. “However” repeats in line 455 and 456. I would suggest changing it.
    2. We followed your suggestion and replaced the word “however” in one of these lines with the word “nonetheless”.
    1. Lines 579 and 584: “RETG810” should be changed for “RET G810”.
    2. We have corrected “RETG810” for “RET G810” in lines 579 and 584. Thank you.
    1. Line 644: “in in 25” should be corrected
    2. We corrected this mistake. Thank you.

Dear Reviewer, thank you for your suggestions. We see that including them in our article made our paper more clear, more understandable and legible for the readers. Thank you.

Sincerely,

The authors.

Reviewer 2 Report

In this extensive review, the authors reported the relevant issues about the management of MTC. However several details are lacking, several references should be included and several concepts should be better rephrased.

Major comments

1.5 Genetics

The authors referred that “Specific for the sporadic form of MTC are mutations involving codons C630 and A883, as well as H568 and S1024”. However, germline mutations were described both at codon C630 and A883 (Refer to Romei, Nature Endocrinology reviews, 2016, doi: 10.1038/nrendo.2016.11). The authors should better clarify this point. 

The authors reported GEP-NENs as a component of MEN2b. This is a relatively new information, could the authors cite some reference about? The used references do not report this data. 

The authors reported “These syndromes are also caused by mutations in the RET gene, but they occur in germline cells”. However, not all cases of MEN2 are caused by “de novo” mutations. The authors should highlight this important point. 

1.6 Prognosis

The author stated that several factors influenced the prognosis. However they do not report any reference about. Please report references about the biomarkers (PMID 33234054, 14715844, 23093698, 18230832), histology (12727956, and guidelines to take other references 25810047), age (33396890, 33974051).

The authors reported that “MTCs are associated with a higher mortality rate than other proliferative thyroid lesions”. However, both poorly differentiated thyroid cancer and anaplastic thyroid cancer has poorer prognosis. The authors should refer only to well differentiated thyroid cancer.

1.8. Laboratory Diagnostics 

The authors reported that “When suspicious nodules are visualized on thyroid ultrasound, FNA biopsy is indicated”. However, recently, Matrone et al. showed that the five main ultrasound (US) risk stratification systems correctly identify less than 50% of MTC (doi: 10.1530/EJE-21-0313); likewise, also other authors reported the same problem (doi: https://doi.org/10.1507/endocrj.ej12-0050). Accordingly, the authors should include this problem in their assumption. 

About CEA, it is not clear if the authors are proposing to use CEA in diagnosis setting or only in monitoring the progression of confirmed disease and in staging. Please clarify.

About RET genetic testing, the authors should clarify if they are talking about germline or somatic mutations. Furthermore, it is could be useful to add that in case of presence of germline RET mutations, all first-degree relatives must be test for RET germline mutations.

The authors should report the recent literature about the use of cell free DNA in MTC diagnosis and monitoring. 

Section 3, please correct Systematic in Systemic

3.1. Multikinase Inhibitors 

The authors should explicit abbreviation of RET at its first use. 

The authors should cite the original article of the phase II study about the use of lenvatinib in patients with unresectable MTC. Recently, other authors reported its use as salvage therapy in patients with advanced MTC (doi: 10.1007/s40618-020-01491-3). 

About anlotinib, the trial whom the authors cited has been recently published (doi: 10.1158/1078-0432.CCR-20-2950) and the authors should report these results. 

About the lack of improvement of OS with MKIs, the authors should report that in both phase III trial evaluating vandetanib and cabozantinib OS was a secondary end-point. Furthermore, it is worth noting that MKIs treatment is not curative, since they have a cytostatic instead of cytotoxic activity. However, from an oncologic point of view stable disease in patients with advanced and progressing neoplastic disease is a significant goal of the treatment. Accordingly, the authors should better clarify the following phrase “There is also no evidence that TKI therapy has a curative effect, as a significant improvement in OS has not been achieved”. 

3.2. Highly Selective RET Inhibitors 

It is not clear at which time point selpercatinib induce a PFS of 82%. 

The data about pralsetinib phase I-II trial has been recently published. The authors should talk about it. 

A relevant point about the use of selpercatinib and pralsetinib in MTC therapy is their interesting toxicity profiles. The authors should talk about it. 

The authors reported the emergence of resistance mechanisms against pralsetinib and selpercatinib. However, it is not clear if these mechanisms were observed in MTC or in other tumors. The authors should distinguish them. 

Author Response

Journal: Cancers                                                                                 July 18, 2022

MDPI

Special Issue: “Advances in Thyroid Cancers”

Dear Editors: Prof. Stefania Masone, Prof. Nunzio Velotti

Dear Editors and Reviewers,

At the very beginning we would like to thank you very much for the possibility to re-submit our revised manuscript entitled Update on the Diagnosis and Management of Medullary Thyroid Cancer: What has Changed in Recent Years?” Thank you very much for considering it for potential publication in Cancers.

We would like to thank the reviewers for the very thorough reviews and for the advices and constructive criticism, which have been valuable for improving our paper. All of the suggestions for changes and improvements were very helpful to us, and we have revised the manuscript according to the recommendations made in the reviews. All of the changed, deleted and added portions of the manuscript are marked by using Track Changes. According to the reviewers’ instructions we corrected our manuscript point-by-point as follows.

Reviewer 2:

Dear Reviewer, thank you very much for your review and suggestions. Thank you very much for the statement, that “the authors reported the relevant issues about the management of MTC”, it is wonderful and important opinion for us. Thank you.  As far as your suggestions, we introduced all of them to the manuscript as follows:

    1. 5 Genetics.
      1. The authors referred that “Specific for the sporadic form of MTC are mutations involving codons C630 and A883, as well as H568 and S1024”. However, germline mutations were described both at codon C630 and A883 (Refer to Romei, Nature Endocrinology reviews, 2016, doi: 10.1038/nrendo.2016.11). The authors should better clarify this point.
      2. The authors reported GEP-NENs as a component of MEN2b. This is a relatively new information, could the authors cite some reference about? The used references do not report this data.
  • The authors reported “These syndromes are also caused by mutations in the RET gene, but they occur in germline cells”. However, not all cases of MEN2 are caused by “de novo” mutations. The authors should highlight this important point.
    1. Thank you very much for this comment. We appreciate that you suggested us a reliable source of knowledge. Some amendments were included in our paper (doi: 10.1038/nrendo.2016.11) and we hope, that now the information included in the genetics section is clear and free from inaccuracies. Thank you.
    2. We realized that we misunderstood the information in used references. Indeed, GEP-NENs are not typical components of MEN2b, so we removed the fragment about it. Thank you for indicating this problem.
  • Thank you for noticing this imprecision. We added a fragment which clarifies the question of the etiology of MEN2 syndrome. We referred to a paper that describes this topic (doi: 10.3390/genes10090698).
    1. 6 Prognosis
      1. The author stated that several factors influenced the prognosis. However they do not report any reference about. Please report references about the biomarkers (PMID 33234054, 14715844, 23093698, 18230832), histology (12727956, and guidelines to take other references 25810047), age (33396890, 33974051).
      2. The authors reported that “MTCs are associated with a higher mortality rate than other proliferative thyroid lesions”. However, both poorly differentiated thyroid cancer and anaplastic thyroid cancer has poorer prognosis. The authors should refer only to well differentiated thyroid cancer.
      1. Thank you very much for this advice. Indeed, more detailed comments about each prognostic factor were needed. We expanded the topic and added more necessary information to the prognosis section. We referred to several necessary papers (PMD 33234054, 14715844, 23093698, 18230832, 33396890, 33974051 and 22946486). Thank you very much.
      2. Thank you for highlighting this point. We changed the comparison object into only well differentiated thyroid cancers.
      1. 8. Laboratory Diagnostics
        1. The authors reported that “When suspicious nodules are visualized on thyroid ultrasound, FNA biopsy is indicated”. However, recently, Matrone et al. showed that the five main ultrasound (US) risk stratification systems correctly identify less than 50% of MTC (doi: 10.1530/EJE-21-0313); likewise, also other authors reported the same problem (doi: https://doi.org/10.1507/endocrj.ej12-0050). Accordingly, the authors should include this problem in their assumption.
        2. About CEA, it is not clear if the authors are proposing to use CEA in diagnosis setting or only in monitoring the progression of confirmed disease and in staging. Please clarify.
  • About RET genetic testing, the authors should clarify if they are talking about germline or somatic mutations. Furthermore, it is could be useful to add that in case of presence of germline RET mutations, all first-degree relatives must be test for RET germline mutations.
  1. The authors should report the recent literature about the use of cell free DNA in MTC diagnosis and monitoring.
    1. Thank you very much, and we are grateful that you noticed this inaccuracy. We added some recent information, that highlight the risk that comes with relying too much on US. We referred to a paper that rises up this problem (doi: 10.1111/cen.14739).
    2. Thank you very much for this comment. We decided to paraphrase the fragment about CEA to make it more clear. We emphasize the usefulness of serum CEA measurement in monitoring the progression of confirmed disease and the staging. Thank you.
  • Indeed, the paragraph about RET genetic testing lacked some important information. We followed your suggestion and we expanded the topic of genetic counseling (doi: 10.3390/genes10090698). Thank you for your advice.
  1. Thank you for this advice. We put more emphasis on the liquid biopsy topic, especially on cfDNA testing, as was suggested. We referred to some recent work about the utility of liquid biopsies (doi: 10.3390/cancers14082028, 10.3390/ijms22147707, 10.1210/jc.2017-01039). We hope that now our paper more reliably describes the novelties of laboratory diagnostics. Thank you.
    1. Section 3, please correct Systematic in Systemic
    2. We corrected the word “Systematic” to “Systemic” in Section 3. Thank you.
    1. 1. Multikinase Inhibitors
      1. The authors should explicit abbreviation of RET at its first use.
      2. The authors should cite the original article of the phase II study about the use of lenvatinib in patients with unresectable MTC. Recently, other authors reported its use as salvage therapy in patients with advanced MTC (doi: 10.1007/s40618-020-01491-3).
  • About anlotinib, the trial whom the authors cited has been recently published (doi: 10.1158/1078-0432.CCR-20-2950) and the authors should report these results.
  1. About the lack of improvement of OS with MKIs, the authors should report that in both phase III trial evaluating vandetanib and cabozantinib OS was a secondary end-point. Furthermore, it is worth noting that MKIs treatment is not curative, since they have a cytostatic instead of cytotoxic activity. However, from an oncologic point of view stable disease in patients with advanced and progressing neoplastic disease is a significant goal of the treatment. Accordingly, the authors should better clarify the following phrase “There is also no evidence that TKI therapy has a curative effect, as a significant improvement in OS has not been achieved”.
    1. We explained RET abbreviation when first used in the text as you rightly suggested. Thank you.
    2. According to your comment, we cited the original article of the phase II study about the use of lenvatinib in patients with unresectable MTC (doi: 1002/cncr.29395). We are grateful for additional information about the recently published paper (doi: 10.1007/s40618-020-01491-3). We have included information about it in the article.
  • Thank you for this information. In our work we tried to use the most recent sources of information available to us, where we looked for diagnostic and therapeutic options. We have updated the information about this study after your comment and we have reported the results. Thank you.
  1. Thank you for this important note. Mentioned by you, phrase is indeed not clear without additional sentences of explanation. The main purpose of this part of our article was to outline the need for examining more specific treatment options, that were described below, in further paragraphs. After a thorough review of this paragraph we decided to remove discussed phrase and elaborate on it in the paragraph above. We also have added some references to support this data. Thank you for your comment.
    1. 2. Highly Selective RET Inhibitors
      1. It is not clear at which time point selpercatinib induce a PFS of 82%.
      2. The data about pralsetinib phase I-II trial has been recently published. The authors should talk about it.
  • A relevant point about the use of selpercatinib and pralsetinib in MTC therapy is their interesting toxicity profiles. The authors should talk about it.
  1. The authors reported the emergence of resistance mechanisms against pralsetinib and selpercatinib. However, it is not clear if these mechanisms were observed in MTC or in other tumors. The authors should distinguish them.
    1. According to more accurate source for this part of the article selpercatinib induce a one year PFS of 82%. We clarified this information in the text. After careful review of this paragraph we also moved the reference to support this data. Thank you.
    2. We appreciate this information. We discussed the data as you suggested.
  • We paid more attention to toxicity profile of selpercatinib and pralsetinib thanks to your comment. Through our mistake the citatation was misattributed. We replaced it with the article by Bartz-Kurycki et al., 2021 DOI: 10.1177/ 20420188211049611.
  1. Indeed we should distinguish what type of tumor was involved in the resistance mechanism. According to citation used by us, the resistance mechanism refers to acquiring mutations in RET G810 in patient with NSCLC and in RET-mutant MTC progressing on selpercatinib. We added this corrected information in the article. We decided to describe in more detailed theory of the resistance mechanism in this paragraphe.

            Dear Reviewer, thank you for your suggestions. We see, that adding this information increased the scientific value of our article. Thank you.

Sincerely,

The authors.

Round 2

Reviewer 2 Report

I thank the authors for the extensive revision of the paper addressing all my suggestions.